# GTPBP1 resolves paused ribosomes to maintain neuronal homeostasis

**Markus Terrey[1,2], Scott I Adamson[3,4], Alana L Gibson[1], Tianda Deng[5], Ryuta Ishimura[6], Jeffrey H Chuang[3], Susan L Ackerman[1]\***

[1]Howard Hughes Medical Institute, Department of Cellular and Molecular Medicine, Section of Neurobiology, Division of Biological Sciences, University of California, San Diego, San Diego, United States; [2]Graduate School of Biomedical Sciences and Engineering, University of Maine, Orono, United States; [3]The Jackson Laboratory for Genomic Medicine, Farmington, United States; [4]Department of Genetics and Genome Sciences, Institute for Systems Genomics, UConn Health, Farmington, United States; [5]Division of Biological Sciences, Section of Molecular Biology, University of California, San Diego, San Diego, United States; [6]The Jackson Laboratory for Mammalian Genetics, Bar Harbor, United States

**Abstract** Ribosome-associated quality control pathways respond to defects in translational elongation to recycle arrested ribosomes and degrade aberrant polypeptides and mRNAs. Loss of a tRNA gene leads to ribosomal pausing that is resolved by the translational GTPase GTPBP2, and in its absence causes neuron death. Here, we show that loss of the homologous protein GTPBP1 during tRNA deficiency in the mouse brain also leads to codon-specific ribosome pausing and neurodegeneration, suggesting that these non-redundant GTPases function in the same pathway to mitigate ribosome pausing. As observed in $Gtpbp2^{-/-}$ mice (Ishimura et al., 2016), GCN2-mediated activation of the integrated stress response (ISR) was apparent in the $Gtpbp1^{-/-}$ brain. We observed decreased mTORC1 signaling which increased neuronal death, whereas ISR activation was neuroprotective. Our data demonstrate that GTPBP1 functions as an important quality control mechanism during translation elongation and suggest that translational signaling pathways intricately interact to regulate neuronal homeostasis during defective elongation.

**\*For correspondence:** sackerman@ucsd.edu

## Introduction

Translation of mRNA into protein utilizes four principle cycles: translation initiation, elongation, termination and ribosome recycling (*Schuller and Green, 2018*). During translation initiation, eukaryotic translation initiation factors (eIFs) mediate assembly of an 80S ribosome and an initiator methionyl-tRNA at the start codon. Subsequently, the elongating 80S ribosome moves along the mRNA decoding mRNA triplets until the termination codon is reached where eukaryotic peptide chain release factors (eRFs) stimulate the release of the nascent protein. Lastly, to permit engagement of ribosomes in new rounds of translation, the terminated 80S ribosome complex is separated into the 40S and 60S subunits and released from the mRNA by ATP-binding cassette sub-family E member 1 (ABCE1).

Although much attention has been dedicated to the regulation of protein synthesis through translation initiation, collective evidence highlights the effect of translation elongation and its kinetics on protein synthesis (*Stein and Frydman, 2019*). Elongation rates not only vary between mRNAs generated from different genes but rates also fluctuate across a given transcript (*Chaney and Clark, 2015*; *Kaiser and Liu, 2018*; *Rodnina, 2016*). Multiple factors including secondary structure of mRNAs, availability of tRNAs, interactions of the nascent peptide with the ribosome and codon identity influence elongation rates, and variations in these rates affect co-translational protein folding,

translational fidelity, and gene expression through mRNA decay (*Brule and Grayhack, 2017*; *Buhr et al., 2016*; *Chaney and Clark, 2015*; *Drummond and Wilke, 2008*; *Rodnina, 2016*; *Spencer et al., 2012*; *Thommen et al., 2017*; *Wolf and Grayhack, 2015*; *Yu et al., 2015*).

With growing evidence suggesting translation elongation is regulated, elaborate mRNA surveillance mechanisms that resolve translation elongation defects have been identified (*Brandman and Hegde, 2016*; *Joazeiro, 2019*). We previously reported that an ENU-induced null mutation (*nmf205*) in the translational GTPase (trGTPase) GTPBP2 causes early, widespread neurodegeneration when present on the common inbred mouse strain C57BL/6J (B6J) (*Ishimura et al., 2014*). Neuron death in B6J-*Gtpbp2^{nmf205/nmf205}* (B6J-*Gtpbp2^{-/-}*) mice is a result of the epistatic interaction between the null mutation in *Gtpbp2* and a hypomorphic mutation present in the B6J strain that disrupts processing of the brain-specific, nuclear encoded tRNA, *n-Tr20*. *n-Tr20* is widely expressed in neurons and is the most highly expressed of the five tRNA$^{Arg}_{UCU}$ genes in the brain, and the B6J-associated mutation reduces the pool of available tRNA$^{Arg}_{UCU}$ (*Ishimura et al., 2014*; *Kapur et al., 2020*). In the absence of *Gtpbp2*, translating ribosomes in the cerebellum exhibit prolonged pauses at AGA codons, suggesting GTPBP2 acts as a ribosome-rescue factor to resolve codon-specific ribosome pausing that occurs when the available pool of cognate tRNAs is limited. The essential role of *Gtpbp2* in neuronal homeostasis was further revealed by the identification of mutations in *Gtpbp2* causing neurological defects and intellectual disabilities in humans (*Bertoli-Avella et al., 2018*; *Carter et al., 2019*; *Jaberi et al., 2016*).

Defects in translation elongation may feedback to regulate translation initiation, supporting the emerging link between translation elongation and initiation to control global translation (*Chu et al., 2014*; *Inglis et al., 2019*; *Liakath-Ali et al., 2018*; *Sanchez et al., 2019*). Translation is highly regulated by two signaling pathways: the mTOR signaling pathway and the integrated stress response (ISR). mTOR complex 1 (mTORC1) phosphorylates the ribosomal protein S6 kinase (S6K1) and the eIF4E binding protein 1 (4E-BP1) to positively regulate translation initiation and elongation (*Nandagopal and Roux, 2015*; *Thoreen, 2017*). The ISR reprograms the translatome by inhibiting translation initiation to suppress global translation via phosphorylation of the translation initiation factor eIF2α (p-eIF2α$^{S51}$) which inhibits formation of the ternary complex, while allowing translation of stress-responsive genes such as ATF4 (*Harding et al., 2003*; *Wortel et al., 2017*). In the B6J-*Gtpbp2^{-/-}* cerebellum, ribosome stalling is accompanied by induction of the ISR (*Ishimura et al., 2014*). Phosphorylation of eIF2α in the B6J-*Gtpbp2^{-/-}* cerebellum is mediated by GCN2, one of four known kinases in mammals (*Dalton et al., 2012*), which are activated during distinct cellular or environmental stressors. Deletion of *Gcn2* in B6J-*Gtpbp2^{-/-}* mice led to increased neurodegeneration, demonstrating GCN2-dependent activation of the ISR acts to partially restore cellular homeostasis (*Chesnokova et al., 2017*; *Dalton et al., 2012*; *Ishimura et al., 2016*).

In addition to *Gtpbp2*, the genome of many eukaryotes contains a related gene, *Gtpbp1* (*Atkinson, 2015*). A previous report demonstrated that loss of *Gtpbp1* in mice did not lead to overt defects, suggesting functional redundancy between *Gtpbp1* and *Gtpbp2* (*Senju et al., 2000*). However, here we demonstrate that loss of *Gtpbp1* in mice with the B6J-associated mutation in *n-Tr20* causes neurodegeneration identical to that observed in B6J-*Gtpbp2^{-/-}* mice. Furthermore, ribosome footprint profiling analysis suggests GTPBP1 functions as a novel ribosome-rescue factor to resolve ribosome pausing defects during tRNA deficiency. As observed in B6J-*Gtpbp2^{-/-}* mice, the ISR is also induced in the B6J.*Gtpbp1^{-/-}* brain and protects neurons from ribosome-pausing induced neurodegeneration. Finally, we show that deficiencies in ribosome pause resolution alter mTOR signaling in a cell type-specific manner, suggesting that differences in the modulation of translational signaling pathways may contribute to the selective neurodegeneration observed with defects in translation elongation.

## Results

### Loss of *Gtpbp1* leads to widespread neurodegeneration when tRNA is deficient

Previously we demonstrated that loss of the trGTPase GTPBP2 causes progressive neurodegeneration in mice (*Ishimura et al., 2014*). However, mice homozygous for a null mutation in *Gtpbp1*, which encodes a structurally related trGTPase (*Figure 1A*), were reported to have no apparent phenotypes on a mixed genetic background (*Senju et al., 2000*). Both GTPBP2 and GTPBP1 are expressed in many tissues, and in situ hybridization revealed that transcripts of these genes were widely expressed throughout the brain (*Ishimura et al., 2014*; *Senju et al., 2000*; *Figure 1B–D*, *Figure 1—figure supplement 1A and B*). Interestingly, expression of *Gtpbp1* and *Gtpbp2* occurred both in neurons that degenerate in *Gtpbp2*[-/-] mice (e.g. cerebellar granule cells, dentate gyrus (DG) granule cells, neurons in cortical layer IV) and those that do not (e.g. hippocampal pyramidal cells and non-layer IV cortical neurons) suggesting possible functional redundancy between these genes in some cell types.

To investigate if phenotypes in *Gtpbp1*[-/-] mice are dependent on genetic background as observed for B6J-*Gtpbp2*[-/-] mice, we generated congenic B6J.*Gtpbp1*[-/-] mice. Like B6J-*Gtpbp2*[-/-] mice, B6J.*Gtpbp1*[-/-] mice were indistinguishable from littermate controls at 3 weeks of age, developed overt ataxia by 6 weeks, and died by 8 weeks of age. Cerebellar degeneration was also similar to that observed in B6J-*Gtpbp2*[-/-] mice (*Ishimura et al., 2014*; *Figure 1E*). Apoptotic granule cells were observed in caudal lobes of the cerebellum just prior to 4 weeks of age and these neurons progressively died in a posterior to anterior manner. As previously observed in B6J-*Gtpbp2*[-/-] mice, granule cells in the DG, layer IV cortical neurons and multiple neurons in the retina degenerated in B6J.*Gtpbp1*[-/-] mice, but cell death was not observed for other neurons such as hippocampal pyramidal cells or neurons in the other layers of the cortex (*Figure 1—figure supplement 2A–E*). Consistent with the dependency of cerebellar granule cell degeneration on levels of *n-Tr20*, a tRNA$^{Arg}_{UCU}$ gene with a processing mutation in the B6J strain, transgenic expression of wild type *n-Tr20* greatly attenuated degeneration of these neurons as has previously been observed for B6J-*Gtpbp2*[-/-] mice (*Figure 1E*).

To genetically test for compensation between *Gtpbp1* and *Gtpbp2*, we analyzed mice that had mutations in both genes. Neurodegeneration was not observed in mice heterozygous for mutations in both *Gtpbp1* and *Gtpbp2* (*Figure 1F*). Furthermore, the onset, progression and specificity of neurodegeneration were similar between B6J.*Gtpbp1*[-/-]; *Gtpbp2*[-/-], B6J.*Gtpbp1*[-/-] and B6J-*Gtpbp2*[-/-] mice (*Figure 1F–H*). These results suggest that *Gtpbp1* and *Gtpbp2* are functionally distinct and act in a common pathway to mediate cellular homeostasis that is necessary for neuron survival.

### GTPBP1 is a novel ribosome-rescue factor

The genetic interaction between the B6J-derived *n-Tr20* mutation and the loss of *Gtpbp1* suggested that like GTPBP2, GTPBP1 might act as a rescue factor for ribosomes paused at AGA codons. To investigate this possibility, we performed ribosome profiling on cerebella from 3-week-old B6J.*Gtpbp1*[-/-] and B6J mice. As previously observed in the B6J-*Gtpbp2*[-/-] cerebellum, ribosome occupancy dramatically increased in the B6J-*Gtpbp1*[-/-] cerebellum when AGA codons were in the A-site of the ribosome (*Figure 2A and B*, *Figure 2—figure supplement 1A*). Comparison of genes with AGA pauses revealed that approximately 50% of pausing genes were shared between libraries generated from individual B6J.*Gtpbp1*[-/-] mice, and this was also true for B6J.*Gtpbp2*[-/-] mice, supporting that AGA pausing is likely stochastic (*Figure 2C*). Similarly, about 50% of genes with AGA pausing intersected between B6J.*Gtpbp1*[-/-] and B6J-*Gtpbp2*[-/-] mice (*Figure 2—figure supplement 1B*, *Supplementary file 1*). Consistent with stochastic pausing at AGA codons, gene ontology (GO) analysis of genes with pauses revealed enrichment in numerous biological processes and the majority of these processes were enriched in both *Gtpbp1* and *Gtpbp2* (*Figure 2—figure supplement 1C*). Thus, our data suggest that both GTPBP1 and GTPBP2 rescue stochastic ribosome-pausing events when the tRNA$^{Arg}_{UCU}$ pool is reduced.

To determine if transcriptional changes are shared upon loss of *Gtpbp1* and *Gtpbp2*, we performed RNA-sequencing analysis. Comparison of data from the cerebellum of 3-week-old B6J.*Gtpbp1*[-/-] and B6J mice revealed significant (q-value $\leq$0.05) differential expression (DE) of

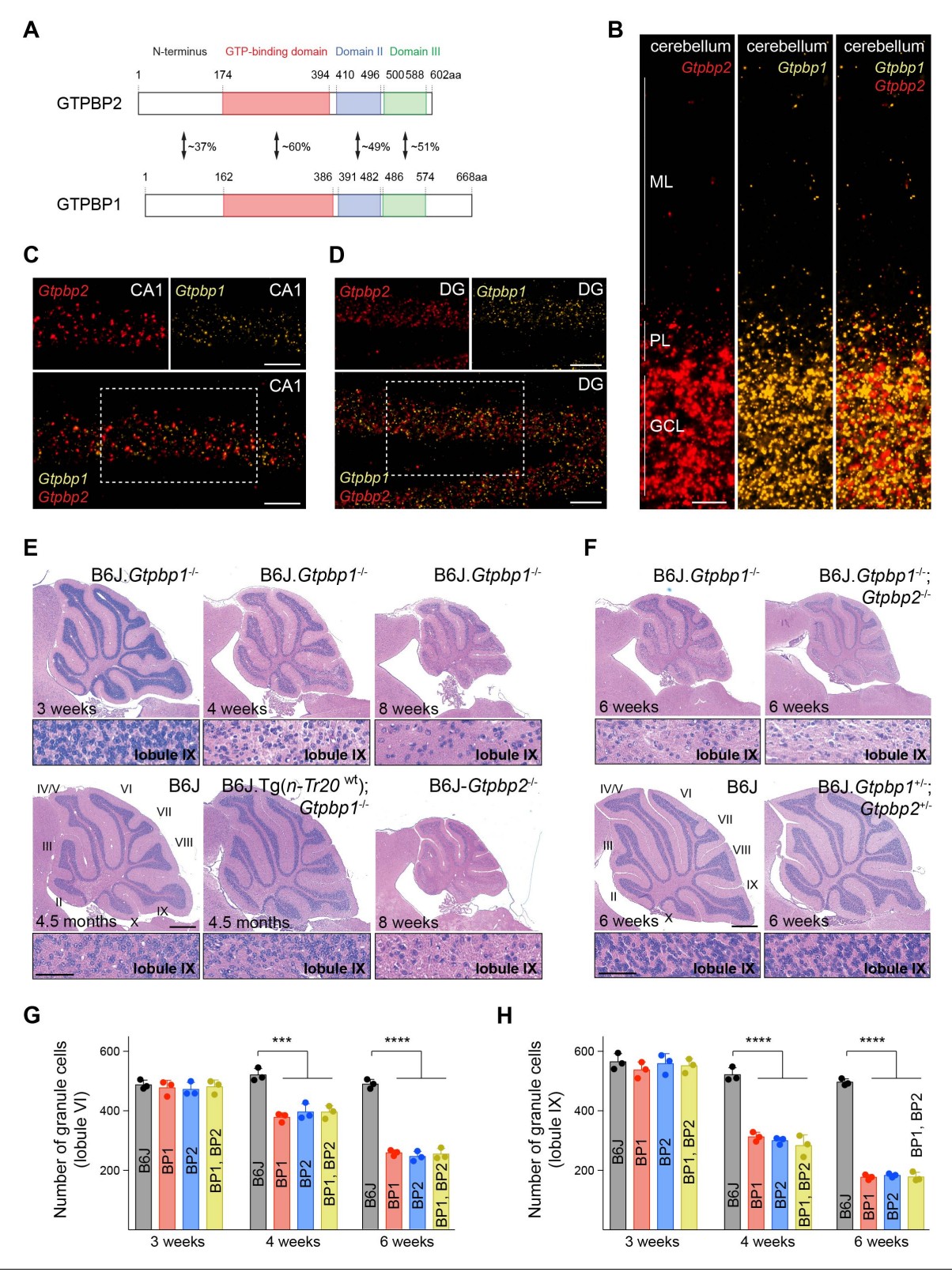

**Figure 1.** tRNA deficiency induces neurodegeneration in B6J.*Gtpbp1*[-/-] mice. (**A**) Domain structure of mouse GTPBP2 and GTPBP1. The percent of identical amino acids for each domain is shown. (**B–D**) In situ hybridization demonstrating ubiquitous expression of *Gtpbp1* (yellow) and *Gtpbp2* (red) in the P28 wild type (B6J) cerebellum (**B**), and CA1 region of the hippocampus (**C**), dentate gyrus (DG) (**D**) (n = 2 mice). Images of individual probes from areas defined by rectangles in *C* and *D* are shown above merged images. (**E, F**) Hematoxylin and eosin staining of sagittal sections of the cerebellum

*Figure 1 continued on next page*

Figure 1 continued

(n = 3–4 mice/genotype). Higher magnification images of lobule IX are shown below each genotype. Cerebellar lobes are indicated by Roman numerals. (G, H) Number of cerebellar granule cells in lobule VI (G) and lobule IX (H) of 3-, 4-, and 6-week-old BP1 (B6J.*Gtpbp1*$^{-/-}$); BP2 (B6J-*Gtpbp2*$^{-/-}$); and BP1, BP2 (B6J.*Gtpbp1*$^{-/-}$; *Gtpbp2*$^{-/-}$) mice (n = 3 mice/genotype). Data represent mean + SD. ML, molecular cell layer; PL, Purkinje cell layer; GCL, granule cell layer. Scale bars: 20 µm (B, C, D); 500 µm and 50 µm (higher magnification) (E, F). One-way ANOVA was corrected for multiple comparisons using Tukey method (G, H). ***p≤0.001, ****p≤0.0001.

The online version of this article includes the following source data and figure supplement(s) for figure 1:

**Source data 1.** tRNA deficiency induces neurodegeneration in B6J.*Gtpbp1*$^{-/-}$ mice.
**Figure supplement 1.** Expression of *Gtpbp1* and *Gtpbp2* in the mouse cortex.
**Figure supplement 2.** Extensive neurodegeneration in B6J.*Gtpbp1*$^{-/-}$ mice.

approximately 12% of the detected genes, with 46% upregulated and 54% downregulated (DE *Gtpbp1*, *Supplementary file 2*). Similarly, loss of *Gtpbp2* altered expression of about 10% of the detected genes with 45% and 55% upregulated and downregulated, respectively, when compared to B6J (DE *Gtpbp2*, *Supplementary file 2*). Interestingly, only 27 genes (mostly non-coding genes) were differentially expressed between *Gtpbp1*$^{-/-}$ and *Gtpbp2*$^{-/-}$ cerebella, revealing that loss of either gene similarly altered the transcriptome (*Figure 2D*, *Supplementary file 2*).

The high similarity of gene expression changes in *Gtpbp1* and *Gtpbp2* mutant mice, combined with our observation that ribosome-pausing defects are likely stochastic, suggested that transcriptional alterations might reflect a common cellular response to ribosome pausing rather than changes in levels of the specific genes that harbor paused ribosomes. In agreement, transcriptional changes were only weakly correlated with genes showing increased ribosomal occupancy (Spearman correlation coefficient of 0.0405, p-value=0.0083). Significant (q-value ≤0.05) changes in expression were only observed in 33% or 26% of the genes with AGA pauses (that were detected in at least two replicates) in the *Gtpbp1*$^{-/-}$ and *Gtpbp2*$^{-/-}$ cerebellum, respectively (*Figure 2E*).

## Loss of GTPBP1 activates the ISR

In order to identify molecular pathways that might respond to ribosome pausing, we performed upstream regulator analysis (Ingenuity Pathway Analysis) of genes differentially expressed between B6J and B6J.*Gtpbp1*$^{-/-}$ mice (DE *Gtpbp1*). This analysis revealed significant enrichment for activation of the transcription factor ATF4, an effector of the ISR (*Figure 3A*, inset). The ISR is induced by the coupling of stress signals to the phosphorylation of serine 51 of the translation initiation factor eIF2α to decrease translation initiation. While the ISR reduces translation of many mRNAs, translation of ATF4 is enhanced, which results in the upregulation of genes to restore cellular homeostasis. We previously observed induction of the ISR in the B6J-*Gtpbp2*$^{-/-}$ cerebellum suggesting that activation of this pathway may be a common cellular response to ribosome stalling (*Ishimura et al., 2016*). In agreement, *Atf4* and 141 of the 153 differentially expressed ATF4 target genes (*Han et al., 2013*) were upregulated in the cerebellum of 3-week-old B6J.*Gtpbp1*$^{-/-}$ mice (DE *Gtpbp1*, *Figure 3A*). Furthermore, levels of phosphorylated eIF2α (p-eIF2α$^{S51}$) were increased in the cerebellum and hippocampus of B6J.*Gtpbp1*$^{-/-}$ mice (*Figure 3B and C*). In agreement, in situ hybridization of the B6J.*Gtpbp1*$^{-/-}$ hippocampus with probes to ATF4 targets induced in the cerebellum of these mice demonstrated that these ATF4 targets were also induced in hippocampal neurons of 3-week-old mutant mice (*Figure 3D*). Interestingly, induction of ATF4 targets varied in different types of neurons. *Sesn2*, *Slc7a1*, and *Chac1* were upregulated in both CA1 and DG neurons, whereas *Ddr2* was only upregulated in CA1 neurons (*Figure 3D*).

In B6J-*Gtpbp2*$^{-/-}$ mice, the ISR is activated by the eIF2α kinase GCN2 (EIF2AK4). Inhibition of ISR activation via *Gcn2* deletion in these mice accelerated cerebellar granule cell death and induced neurodegeneration of hippocampal pyramidal cells (*Ishimura et al., 2016*). GCN2 was enriched in upstream regulator analysis of differentially expressed genes in the B6J.*Gtpbp1*$^{-/-}$ cerebellum (DE *Gtpbp1*), suggesting that this kinase may also mediate activation of the ISR in *Gtpbp1*-deficient mice (*Figure 3A*, inset). In agreement, *Gcn2* deletion in B6J.*Gtpbp1*$^{-/-}$ mice accelerated degeneration of cerebellar granule cells. In 5-week-old B6J.*Gtpbp1*$^{-/-}$; *Gcn2*$^{-/-}$ mice, 65% of these neurons had degenerated compared to 37% in B6J.*Gtpbp1*$^{-/-}$ mice (*Figure 3E and F*). Furthermore, the DG of B6J.*Gtpbp1*$^{-/-}$; *Gcn2*$^{-/-}$ mice had twice the number of granule cells with pyknotic nuclei compared to B6J.*Gtpbp1*$^{-/-}$ mice (*Figure 3G*, *Figure 3—figure supplement 1*). Finally, although CA1 pyramidal

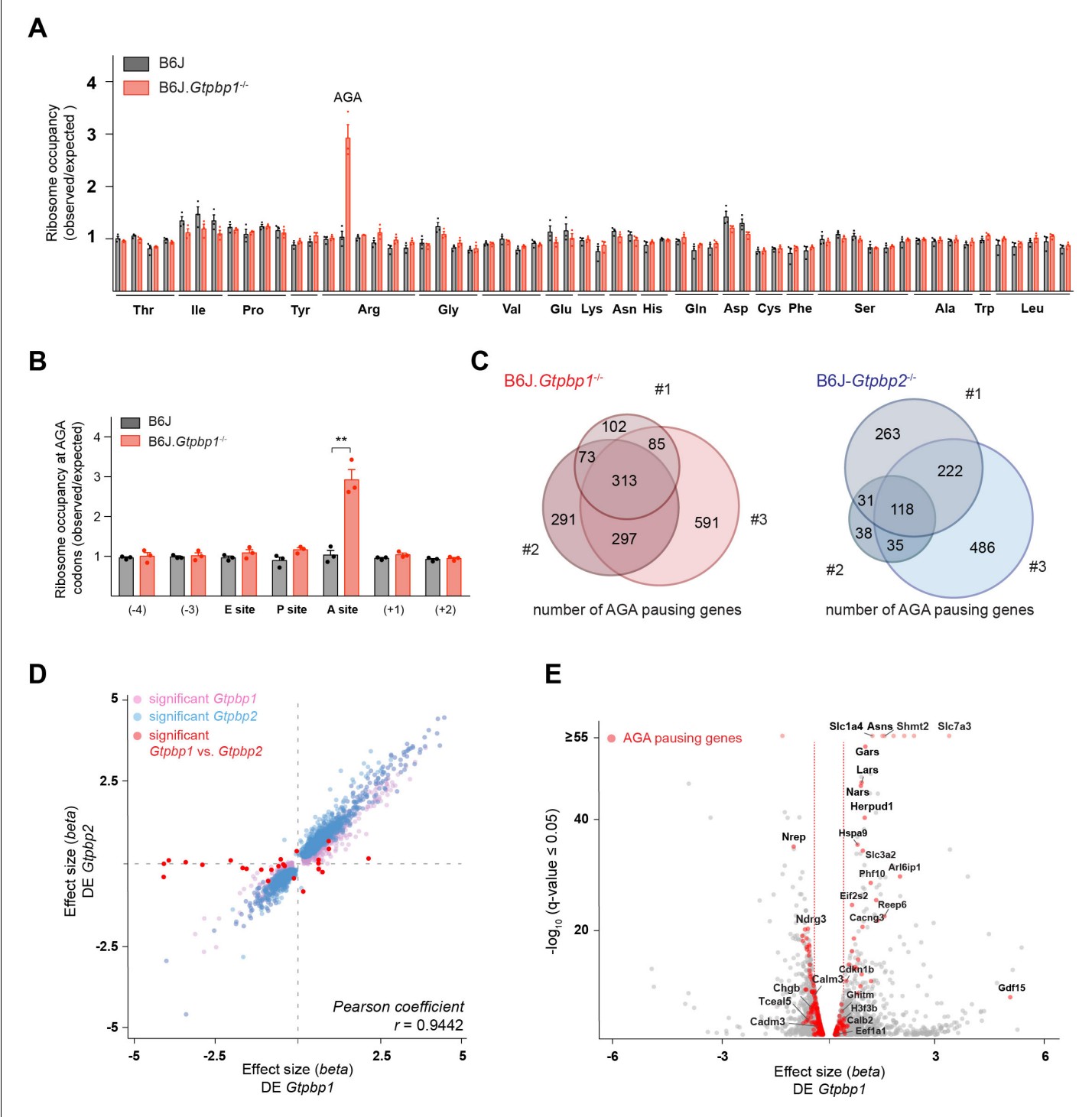

**Figure 2.** GTPBP1 resolves ribosome pausing induced by tRNA deficiency. (**A**) Ribosome occupancy was calculated by dividing the number of genome-wide reads at codons by the expected reads in the ribosomal A-site (n = 3 biological replicates). Data represent mean + SEM. Note that ribosome occupancy increased only at AGA codons in P21 B6J.*Gtpbp1*⁻/⁻ mice. (**B**) Ribosome occupancy at AGA codons was calculated by dividing genome-wide reads at AGA codons by expected reads. Data represent mean + SEM from ribosome profiling from cerebella of P21 B6J and B6J.*Gtpbp1*⁻/⁻ mice (n = 3 biological replicates). (**C**) Venn diagram of genes with increased ribosome occupancy at AGA codons (z-score ≥10) between libraries prepared from individual B6J.*Gtpbp1*⁻/⁻ (red) or B6J-*Gtpbp2*⁻/⁻ (blue) mice (biological replicates 1–3). (**D**) Significant (q-value ≤0.05, q-value refers to the corrected p-value using Benjamini-Hochberg correction) transcriptional changes in gene expression between P21 B6J and B6J.*Gtpbp1*⁻/⁻ cerebellum (x-axis, DE *Gtpbp1*) are plotted against those between P21 B6J and B6J-*Gtpbp2*⁻/⁻ cerebellum (y-axis, DE *Gtpbp2*) (n = 3 biological replicates). Significant (q-

*Figure 2 continued on next page*

Figure 2 continued

value ≤0.05) *Gtpbp1*-dependent expression changes are colored in pink and *Gtpbp2*-dependent expression changes in blue. Significant (q-value ≤0.05) transcriptional changes in gene expression between B6J.*Gtpbp1*$^{-/-}$ and B6J-*Gtpbp2*$^{-/-}$ are shown in red (27 genes). The *beta* effect size is analogous to the natural log fold change in expression. (E) Analysis of differential gene expression between P21 B6J and B6J.*Gtpbp1*$^{-/-}$ mice (DE *Gtpbp1*). Significant (q-value ≤0.05) transcriptional changes in gene expression are shown in grey, and genes with increased AGA ribosome occupancy (z-score ≥10, detected in at least two biological replicates) that are differentially expressed are highlighted in red (260 genes). The *beta* effect size is analogous to the natural log fold change in expression and 1.5-fold changes in gene expression are indicated by the red dashed lines. Multiple *t* tests were corrected for multiple comparisons using Holm-Sidak method (B). **p≤0.01.

The online version of this article includes the following source data and figure supplement(s) for figure 2:

**Source data 1.** GTPBP1 resolves ribosome pausing induced by tRNA deficiency.
**Figure supplement 1.** *Gtpbp1* and *Gtpbp2* resolve AGA pauses.

neurons do not normally degenerate in B6J.*Gtpbp1*$^{-/-}$ mice, 10.3% of these neurons were undergoing apoptosis in the hippocampus of 5-week-old B6J.*Gtpbp1*$^{-/-}$; *Gcn2*$^{-/-}$ mice (**Figure 3G and H**). As previously reported (**Ishimura et al., 2016**; **Zhang et al., 2002**), neurodegeneration was not observed in the B6J.*Gcn2*$^{-/-}$ brain (data not shown).

## Loss of *Gtpbp1* or *Gtpbp2* induces neuron-specific changes in mTOR signaling

Our data suggest that defects in translation elongation activate the ISR, which regulates translation initiation and influences neurodegeneration. Thus, we wondered if ribosome pausing in *Gtpbp1* or *Gtpbp2* mutant mice could alter additional translational control pathways. Upstream regulator analysis of differentially expressed genes in the cerebellum of B6J.*Gtpbp1*$^{-/-}$ mice (DE *Gtpbp1*) revealed enrichment, although lower than that observed for ATF4, of the mammalian target of rapamycin kinase (mTOR) (**Figure 3A**, inset). mTOR functions in two distinct protein complexes known as mTOR complex 1 (mTORC1) and complex 2 (mTORC2). Although association with membrane bound ribosomes (i.e. the endoplasmic reticulum) has been observed for mTORC2, the central role of mTOR in protein translation is largely attributed to mTORC1 through phosphorylation of specific effector proteins and translation of particular genes that contain 5' terminal oligopyrimidine (TOP) tracts (**Dai and Lu, 2009**; **Laplante and Sabatini, 2013**; **Laplante and Sabatini, 2009**; **Ma and Blenis, 2009a**; **Ma and Blenis, 2009b**; **Nandagopal and Roux, 2015**; **Zinzalla et al., 2011**). To determine if mTOR activity is altered by loss of *Gtpbp1* or *Gtpbp2*, we analyzed the translational efficiency (TE) (**Supplementary file 3**) of genes regulated by mTORC1 via their TOP motifs by normalizing the number of ribosome footprint reads to that of RNA-sequencing reads. In addition, we assessed the phosphorylation status of the ribosomal protein S6 (p-S6$^{240/244}$), a known downstream target of mTORC1 (**Saxton and Sabatini, 2017**; **Thoreen et al., 2012**). Only one of the 53 detected 5'TOP genes had an altered TE in the B6J.*Gtpbp1*$^{-/-}$ cerebellum (*Rps29*, TE *Gtpbp1*) or in the B6J-*Gtpbp2*$^{-/-}$ cerebellum (*Rplp0*, TE *Gtpbp2*) compared to the B6J cerebellum (**Supplementary file 3**, **Figure 4—figure supplement 1A**). In addition, levels of p-S6$^{240/244}$ were unchanged at 3 weeks of age in the cerebellum of both mutant strains, suggesting that mTOR signaling is not altered by ribosome pausing in granule cells, which comprise the vast majority of cells in the cerebellum (**Figure 3B and C**).

In contrast to the cerebellum, p-S6$^{240/244}$ levels were decreased by about 90% in the hippocampus of 3-week-old B6J.*Gtpbp1*$^{-/-}$ and B6J-*Gtpbp2*$^{-/-}$ mice relative to wild type (**Figure 3B and C**). Interestingly, immunofluorescence using antibodies to p-S6$^{240/244}$ revealed that the reduction of p-S6$^{240/244}$ in the mutant hippocampus was more pronounced in specific neuron populations. In the control hippocampus, p-S6$^{240/244}$ signal was most intense in CA3 pyramidal cells and lower in CA1 pyramidal cells and DG granule cells (**Figure 4A**). In B6J.*Gtpbp1*$^{-/-}$ and B6J-*Gtpbp2*$^{-/-}$ mutant hippocampi, p-S6$^{240/244}$ levels were dramatically reduced in the granule cells of the DG and some scattered CA1 neurons (**Figure 4A**). However, levels of p-S6$^{240/244}$ were not affected in CA3 or cortical neurons (i.e. layer IV cortical neurons) of 3-week-old B6J.*Gtpbp1*$^{-/-}$ and B6J-*Gtpbp2*$^{-/-}$ mice (**Figure 4—figure supplement 1B and C**). Together, these results demonstrate that changes in mTOR signaling may occur in a celltype-specific manner in the brains of B6J.*Gtpbp1*$^{-/-}$ and B6J-*Gtpbp2*$^{-/-}$ mice.

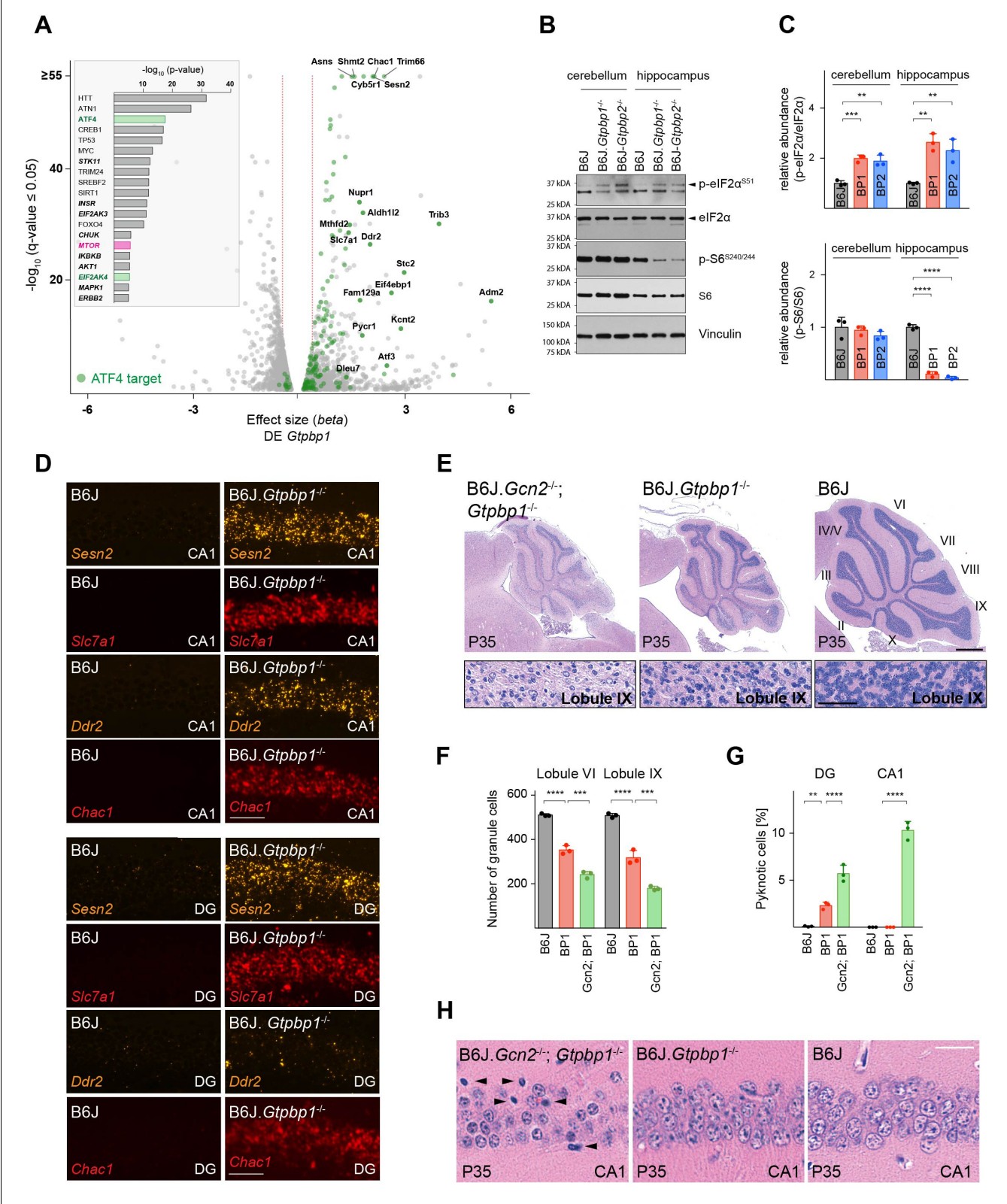

**Figure 3.** Ribosome pausing activates the integrated stress response (ISR) to ameliorate neurodegeneration in *Gtpbp1*[-/-] mice. (**A**) Analysis of transcriptional changes in gene expression between P21 B6J and B6J.*Gtpbp1*[-/-] mice (DE *Gtpbp1*) (n = 3 biological replicates). Significant (q-value ≤0.05, q-value refers to the corrected p-value using Benjamini-Hochberg correction) changes in expression are shown in grey and differentially expressed ATF4 targets are highlighted in green. The *beta* effect size is analogous to the natural log fold change in expression and 1.5-fold changes in

*Figure 3 continued on next page*

Figure 3 continued

gene expression are indicated (red dashed lines). (Inset) Identification of upstream regulators using Ingenuity Pathway Analysis (IPA) of differentially expressed genes between B6J and B6J.Gtpbp1$^{-/-}$ mice (DE Gtpbp1). Top ten transcription factors and kinases (*italics*) are shown. (B) Western blot analysis of tissue lysates from P21 mice. Vinculin was used as an input control. (C) Relative abundance of p-eIF2α$^{S51}$ and p-S6$^{S240/244}$ in the hippocampus and cerebellum of BP1 (B6J.Gtpbp1$^{-/-}$), BP2 (B6J-Gtpbp2$^{-/-}$), and control (B6J) mice (n = 3 mice/genotype). Levels of p-eIF2α$^{S51}$ or p-S6$^{S240/244}$ were normalized to total level of eIF2α or S6, and phosphorylation levels are relative to those of B6J. Data represent mean + SD. (D) In situ hybridization of ATF4 targets in the hippocampal CA1 and dentate gyrus (DG) at P21 (n = 3 mice/genotype). (E) Sagittal cerebellar sections stained with hematoxylin and eosin. Higher magnification images of lobule IX are shown below each genotype (n = 3 mice/genotype). Cerebellar lobes are indicated by Roman numerals. (F) Quantification of cerebellar granule cells of lobule VI and lobule IX of control (B6J), BP1 (B6J.Gtpbp1$^{-/-}$), and Gcn2; BP1 (B6J.Gcn2$^{-/-}$; Gtpbp1$^{-/-}$) mice at P35 (n = 3 mice/genotype). Data represent mean + SD. (G) Percent of DG and CA1 neurons that are pyknotic in control (B6J), BP1 (B6J.Gtpbp1$^{-/-}$), and Gcn2; BP1 (B6J.Gcn2$^{-/-}$; Gtpbp1$^{-/-}$) mice at P35 (n = 3 mice/genotype). Data represent mean + SD. (H) Sagittal sections of the CA1 area of the hippocampus stained with hematoxylin and eosin (n = 3 mice/genotype). Arrowheads indicate pyknotic cells. Scale bars: 50 µm (D); 500 µm and 50 µm (higher magnification) (E); 20 µm (H). One-way ANOVA was corrected for multiple comparisons using Tukey method (C, F, G). **p≤0.01, ***p≤0.001, ****p≤0.0001.

The online version of this article includes the following source data and figure supplement(s) for figure 3:

**Source data 1.** Ribosome pausing activates theintegrated stress response (ISR) to ameliorate neurodegeneration in Gtpbp1-/- mice.
**Figure supplement 1.** Loss of GCN2 enhances hippocampal degeneration in B6J.

The decrease in translation initiation by inhibition of the mTOR pathway has been suggested to reduce ribosome pausing during amino acid deprivation (*Darnell et al., 2018*) suggesting that like the ISR, this pathway may also be protective in the Gtpbp1 and Gtpbp2 mutant brain. However, changes in levels of p-S6$^{240/244}$ in the mutant hippocampus suggested that decreases in mTOR activity were most profound in hippocampal neurons that ultimately degenerate (i.e. the granule cells of the DG). Thus, to determine the role of mTOR signaling on neuron survival during ribosome pausing, we pharmacologically inhibited mTOR by treating B6J, B6J.Gtpbp1$^{-/-}$ and B6J-Gtpbp2$^{-/-}$ mice with rapamycin daily for two weeks beginning at P14. Similar to the dramatic decrease in p-S6$^{240/244}$ in 3-week-old hippocampi in B6J.Gtpbp1$^{-/-}$ and B6J-Gtpbp2$^{-/-}$ mice, cerebellar levels of p-S6$^{240/244}$ were decreased by approximately 85% by P21 in rapamycin-treated mice (*Figure 4—figure supplement 1D–F*). Examination of the cerebellum from P28 B6J.Gtpbp1$^{-/-}$ and B6J-Gtpbp2$^{-/-}$ cerebellum revealed that rapamycin treatment increased granule cell loss by 30% (lobule VI) and 40% (lobule IX) compared to mutant mice treated with vehicle (*Figure 4B and C*). No neuron loss was observed in rapamycin-treated control (B6J) mice (*Figure 4B and C*). Although it has been reported that mTOR inhibition may cause repression of Atf4 transcripts and some of its target genes (*Park et al., 2017*), no significant reduction of Atf4 or its targets was observed in rapamycin-treated mutant mice, suggesting that the ISR and mTOR pathways function independently upon loss of Gtpbp1 and Gtpbp2 in the B6J cerebellum (*Figure 4—figure supplement 1G*).

Rapamycin treatment also increased apoptosis of layer IV cortical neurons by about 3-fold in P28 B6J.Gtpbp1$^{-/-}$ and B6J-Gtpbp2$^{-/-}$ mice but did not cause death of other cortical neurons (*Figure 4—figure supplement 2A and B*). In addition, death of granule cells in the DG was accelerated upon rapamycin treatment. Pyknotic nuclei were observed in 22% of these neurons in P28 rapamycin-treated B6J.Gtpbp1$^{-/-}$ or B6J-Gtpbp2$^{-/-}$ mice, whereas only 3.4% of these neurons had pyknotic nuclei in vehicle-treated mutant mice (*Figure 4—figure supplement 2C and D*). In addition, the number of granule cells was decreased by 35% in the DG of rapamycin-treated mice, indicating that apoptosis in this region of the brain began earlier (*Figure 4—figure supplement 2E*). Finally, apoptosis of CA1 neurons was not observed in rapamycin-treated B6J.Gtpbp1$^{-/-}$ and B6J-Gtpbp2$^{-/-}$ mice at P28 when treatment began at P14. However, pyknotic nuclei were observed when mice were treated from P28-P42, suggesting that while these neurons are sensitive to mTOR suppression, the timing of this sensitivity differs from that of other neurons (*Figure 4—figure supplement 2F*). Taken together, our data demonstrate that unlike the ISR, which acts to prevent loss of neurons with ribosome stalling, inhibition of mTOR increases the vulnerability of multiple neuronal populations.

## Discussion

Ribosome speed during translation is regulated to allow proper folding of the nascent peptide and functional protein production. However, prolonged pausing or stalling of translating ribosomes can

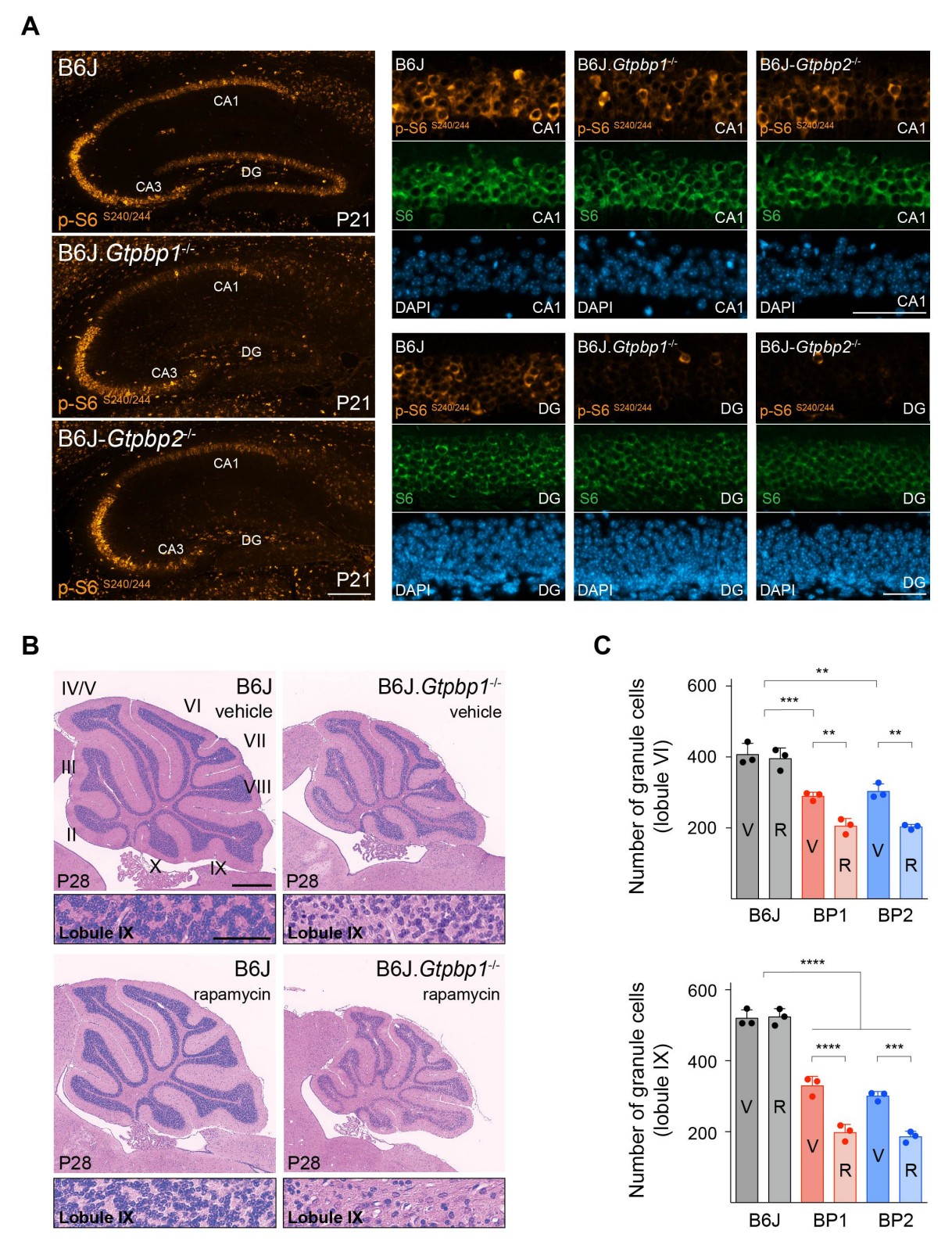

**Figure 4.** Decreased mTOR signaling enhances neurodegeneration in trGTPase-deficient mice. (**A**) Immunofluorescence of P21 hippocampal sections with antibodies against p-S6$^{S240/244}$ (orange) and S6 (green). Sections were counterstained with DAPI (blue). Higher magnifications of CA1 and dentate gyrus (DG) are shown (n = 3 mice/genotype). (**B**) Sagittal cerebellar sections of P28 mice injected with vehicle or rapamycin for 14 days stained with hematoxylin and eosin (n = 3 mice/genotype). Higher magnification images of lobule IX are shown below each genotype. Cerebellar lobes are

*Figure 4 continued on next page*

Figure 4 continued

indicated by Roman numerals. (**C**) Quantification of cerebellar granule cells in lobule VI and lobule IX of either vehicle (**V**) or rapamycin (**R**) treated control (B6J), BP1 (B6J.*Gtpbp1*$^{-/-}$), and BP2 (B6J-*Gtpbp2*$^{-/-}$) mice (n = 3 mice/genotype). Data represent mean + SD. Scale bars: 100 µm and 50 µm (higher magnification) (**A**); 500 µm and 50 µm (higher magnification) (**B**). Two-way ANOVA was corrected for multiple comparisons using Tukey method (**C**). **$p \leq 0.01$, ***$p \leq 0.001$, ****$p \leq 0.0001$.

The online version of this article includes the following source data and figure supplement(s) for figure 4:

**Source data 1.** Decreased mTOR signaling enhances neurodegeneration in trGTPase-deficient mice.

**Figure supplement 1.** Analysis of mTOR signaling in trGTPase-deficient mice.

**Figure supplement 1—source data 1.** Decreased mTOR signaling enhances neurodegeneration in trGTPase-deficient mice.

**Figure supplement 2.** Inhibition of mTOR signaling accelerates cell death in B6J.*Gtpbp1*$^{-/-}$ and B6J-*Gtpbp2*$^{-/-}$ mice.

**Figure supplement 2—source data 1.** Inhibition of mTOR signaling accelerates cell death in B6J.*Gtpbp1*$^{-/-}$ and B6J-*Gtpbp2*$^{-/-}$ mice.

be detrimental to cells. Thus cells have evolved multiple quality control mechanisms that mediate steps from recognition of stalled ribosomes to the resolution of these complexes (*Brandman and Hegde, 2016*; *Joazeiro, 2019*). We previously identified GTPBP2 as a ribosome-rescue factor essential for neuronal survival during tRNA deficiency (*Ishimura et al., 2014*). The high homology of GTPBP1 and GTPBP2, their broad expression patterns, and the lack of overt phenotypes of *Gtpbp1*$^{-/-}$ mice on a mixed genetic background had suggested functional redundancy of these translational GTPases (*Senju et al., 2000*). However, we show here that loss of *Gtpbp1* leads to widespread neurodegeneration in B6J mice. As observed in *Gtpbp2*$^{-/-}$ mice, neuron loss in *Gtpbp1*$^{-/-}$ mice is dependent on a mutation in the B6J strain that disrupts *n-Tr20* tRNA$^{Arg}_{UCU}$ processing (*Ishimura et al., 2014*). Ribosome profiling of the *Gtpbp1*$^{-/-}$ cerebellum revealed increased ribosome occupancy at AGA codons as we previously observed for *Gtpbp2*$^{-/-}$ mice (*Ishimura et al., 2014*). These data suggest that both GTPBP1 and GTPBP2 function as ribosome-rescue factors and are essential genes in many neurons when *n-Tr20* levels are diminished. In addition, the lack of neurodegeneration in mice heterozygous for mutations in both *Gtpbp1* and *Gtpbp2* and the lack of an additive phenotype in the brains of mice lacking both genes suggest that these trGTPases function in the same pathway to mitigate ribosome stalling, perhaps mediating different steps in this pathway.

GTPBP1 and GTPBP2 share domain homology with other trGTPases including the yeast protein Hbs1 (HBS1L in mammals) (*Atkinson, 2015*). Hbs1 and its interacting partner Dom34 recognize paused ribosomes at the 3'end of truncated mRNAs and in the 3'UTR of mRNAs, and the resolution of paused ribosomes is in part gated by the GTPase activity of Hbs1 (*Becker et al., 2012*; *Guydosh et al., 2017*; *Guydosh and Green, 2014*; *Juszkiewicz et al., 2020*; *Pisareva et al., 2011*; *Shoemaker and Green, 2011*). The mechanism by which GTPBP1 and GTPBP2 mitigate defects in ribosome elongation is unknown. Recent biochemical studies revealed that the GTPase activity of GTPBP1 and GTPBP2 is not stimulated in the presence of 80S ribosomes (*Zinoviev et al., 2018*), suggesting that these two trGTPases likely function differently than Hbs1. In agreement, Dom34/ Hbs1 did not mediate pause resolution in a codon-specific manner when pausing was induced at His codons upon inhibition of histidine biosynthesis (*Guydosh and Green, 2014*).

During the process of pause resolution, multiple factors are recruited to mediate ribosome recycling and degradation of the nascent peptide chains or mRNAs associated with stalled ribosomes (*Collart and Weiss, 2020*; *D'Orazio et al., 2019*; *Ibrahim et al., 2018*; *Pelechano et al., 2015*). Interestingly, GTPBP1 can stimulate exosomal degradation of mRNAs (*Woo et al., 2011*), unlike GTPBP2 (*Zinoviev et al., 2018*), and thus may provide a link between elongation defects and mRNA degradation. Although we did not observe a strong correlation between mRNA abundance and ribosome pausing in the *Gtpbp1*$^{-/-}$ cerebellum, the stochastic nature of ribosome stalling could make it difficult to observe the corresponding, and similarly stochastic, changes in mRNA decay. RNA-sequencing techniques that are specifically designed to capture degradation intermediates may be more effective in investigating mRNA degradation in vivo (*Ibrahim et al., 2018*).

As we previously observed in the B6J-*Gtpbp2*$^{-/-}$ cerebellum (*Ishimura et al., 2016*), loss of *Gtpbp1* induced the ISR via activation of GCN2, and deletion of *Gcn2* in B6J.*Gtpbp1*$^{-/-}$ mice resulted in accelerated, and more widespread, neurodegeneration. GCN2 can be activated by uncharged tRNA that accumulates during amino acid deprivation (*Masson, 2019*). However, increases in uncharged tRNA were not observed in the B6J-*Gtpbp2*$^{-/-}$ cerebellum, suggesting other mechanisms underlie GCN2 activation (*Ishimura et al., 2016*). Indeed, recent biochemical and genetic studies

revealed that the P-stalk, a pentameric complex located near the A-site of the ribosome, interacts with GCN2 and activates it, suggesting that translation elongation defects may be sensed directly by GCN2 (*Harding et al., 2019*; *Inglis et al., 2019*). Activation of the ISR and its accompanying protective function were observed across multiple neuronal populations in *Gtpbp1*<sup></sup> and *Gtpbp2*<sup></sup> mice. Interestingly, expression of some ATF4 target genes varied between types of neurons, suggesting that additional cell type-specific mechanisms, such as those controlling formation of ATF4 heterodimers, post-translational modifications of ATF4, or epigenetic regulation of target genes may modulate expression of these genes (*Wortel et al., 2017*).

We also show that defects in ribosome pause resolution in the B6J.*Gtpbp1*<sup>-/-</sup> and B6J-*Gtpbp2*<sup>-/-</sup> brain are associated with decreased mTORC1 signaling. In contrast to ISR activation, which prolonged neuronal survival, inhibition of mTOR signaling negatively affected neuronal survival in mutant mice, despite the fact that the observed changes in both of these pathways are associated with decreased translation initiation. Previous studies showed that during amino acid starvation of HEK293T cells, reduction of mTOR activity and translation correlated with a reduction in ribosome pausing (*Darnell et al., 2018*). While inhibition of mTORC1 may be beneficial at the level of translation in some instances of ribosome stress, the negative impact we observe on neuronal survival may reside in the severity or the duration of the stressor and/or the influence of mTORC1 on other cellular processes including ribosomal RNA processing, gene transcription, protein degradation, and ribosomal and mitochondrial biogenesis (*Laplante and Sabatini, 2013*; *Laplante and Sabatini, 2009*; *López KG de la, 2019*; *Mayer and Grummt, 2006*; *Puertollano, 2014*).

The dramatic downregulation of mTORC1 activity in granule cells of the B6J.*Gtpbp1*<sup>-/-</sup> and B6J-*Gtpbp2*<sup>-/-</sup> DG relative to other neurons suggests that the cellular context plays a role in modulating this signaling pathway upon ribosomal stress. A previous study observed increased mTOR activity in epidermal stem cells upon loss of the ribosome recycling factor *Pelo*, and suppression of mTOR signaling partially restored cellular defects in vivo (*Liakath-Ali et al., 2018*). We recently observed that the relatively low levels of ribosome pausing that occur upon tRNA deficiency (with normal levels of GTPBP1 and GTPBP2) can lead to mTORC1 inhibition and alter neuronal physiology (*Kapur et al., 2020*). Together, these data demonstrate that elongation defects may lead to hyper- or hypoactivity of mTOR signaling and these changes in mTORC1 appear to negatively modulate cellular homeostasis and survival.

Whether mTORC1 responds directly to translational stress or alterations in this signaling pathway are due to changes in cellular homeostasis (e.g. amino acid metabolism, energy levels, or neuronal activity) is unknown, as is the mechanism underlying its cell type-specific response. Proteomic experiments have revealed mTOR-dependent phosphorylation of translation initiation factors and ribosomal proteins. Interestingly, multiple phosphorylation sites were observed on the surface of the 80S ribosome, suggesting that mTOR or mTOR-associated kinases can access the ribosome (*Jiang et al., 2016*). The interaction of mTOR and/or its downstream kinases with the ribosome may provide a mechanism for mTOR to simultaneously monitor and regulate translation. Because protein synthesis varies between cell types and changes during development (*Blair et al., 2017*; *Buszczak et al., 2014*; *Castelo-Szekely et al., 2017*; *Gonzalez et al., 2014*; *Sudmant et al., 2018*), the rate of translation may differentially impact elongation defects and thus the signaling pathways which control translation.

## Materials and methods

**Key resources table**

| Reagent type (species) or resource | Designation | Source or reference | Identifiers | Additional information |
|---|---|---|---|---|
| Strain, strain background (mouse) | *Gtpbp1 < tm1Ynim/tm1Ynim>* | *Senju et al., 2000* | RRID:MGI:3036546 | *Gtpbp1*<sup>-/-</sup> *strain (mixed genetic background)* |
| Strain, strain background (mouse) | *B6J.Gtpbp1<tm1Ynim/tm1Ynim>* | This study | MGI:6467940 | *B6J.Gtpbp1*<sup>-/-</sup> *strain (congenic background, C57BL/6J)* |

*Continued on next page*

*Continued*

| Reagent type (species) or resource | Designation | Source or reference | Identifiers | Additional information |
|---|---|---|---|---|
| Strain, strain background (mouse) | B6J-*Gtpbp2*$^{-/-}$ (C57BL/6J-*Gtpbp2*$^{nmf205}$/J) | *Ishimura et al., 2014* | RRID:IMSR_JAX:004823 | |
| Strain, strain background (mouse) | B6J.*Gcn2*$^{-/-}$ (B6.129S6-*Eif2ak4*$^{tm1.2Dron}$) | The Jackson Laboratory | RRID:IMSR_JAX:008240 | |
| Sequence-based reagent | Genotyping wild type allele Gtpbp1 | This study | N/A | Forward Primer: 5'GAGTACGGGCTGAGTGAAGC3'; Reverse Primer: 5'TGGACAGGAACCTGATGTGA3' |
| Sequence-based reagent | Genotyping mutant allele Gtpbp1 | This study | N/A | Forward Primer: 5'TACGCCACCGTGAAGAGCAT3'; Reverse Primer: 5'AGGGGAGGAGTGGAAGGTGG3' |
| Sequence-based reagent | Quantitative RT-PCR (beta actin) | This study | N/A | Forward Primer: 5'GGCTGTATTCCCCTCCATCG3'; Reverse Primer:5'CCAGTTGGTAACAATGCCATGT3' |
| Sequence-based reagent | RNAscope probe *Gtpbp2* (mouse) | Advanced Cell Diagnostics | #527461 | |
| Sequence-based reagent | RNAscope probe *Gtpbp1* (mouse) | Advanced Cell Diagnostics | #527451-C3 | |
| Sequence-based reagent | RNAscope probe *Sesn2* (mouse) | Advanced Cell Diagnostics | | Probe (reference number: 574751-C2) was modified for this study to be compatible with manual RNAscope protocol but is otherwise equivalent to #574758-C2 |
| Sequence-based reagent | RNAscope probe *Slc7a1* (mouse) | Advanced Cell Diagnostics | #461021 | |
| Sequence-based reagent | RNAscope probe *Ddr2* (mouse) | Advanced Cell Diagnostics | #405991-C2 | |
| Sequence-based reagent | RNAscope probe *Chac1* (mouse) | Advanced Cell Diagnostics | #514501 | |
| Commercial assay or kit | RNAscope Multiplex Fluorescent Reagent Kit v2 | Advanced Cell Diagnostics | #323100 | |
| Commercial assay or kit | TSA Plus Cyanine 5 | PerkinElmer | NEL745001KT | (1:1000) |
| Commercial assay or kit | TSA Plus Cyanine 3 | PerkinElmer | NEL744001KT | (1:2000) |
| Commercial assay or kit | DNA-free DNA Removal Kit | Life Technologies | AM1906 | |
| Commercial assay or kit | SuperScript III First-Strand Synthesis System | Invitrogen | #18080051 | |
| Commercial assay or kit | iQ SYBR Green Supermix | Bio-Rad | #1708880 | |
| Commercial assay or kit | TruSeq v2 mRNA kit | Illumina | RS-122–2001 | |
| Chemical compound, drug | Rapamycin | LC Laboratories | R5000 | |
| Antibody | Rabbit anti-phospho-EIF2alpha (polyclonal) | Cell Signaling Technology | CST #9721; RRID:AB_330951 | WB (1:1000) |
| Antibody | Rabbit anti- EIF2alpha (polyclonal) | Cell Signaling Technology | CST #9722; RRID:AB_2230924 | WB (1:2000) |
| Antibody | Rabbit anti-phospho S6 ribosomal protein, S240/244(polyclonal) | Cell Signaling Technology | CST #5364; RRID:AB_10694233 | WB (1:4000) IF (1:1000) |

*Continued on next page*

*Continued*

| Reagent type (species) or resource | Designation | Source or reference | Identifiers | Additional information |
|---|---|---|---|---|
| Antibody | Mouse anti-S6 ribosomal protein (monoclonal) | Santa Cruz Biotechnology | sc-74459; RRID:AB_1129205 | WB (1:2000) IF (1:500) |
| Antibody | Mouse anti-Vinculin (monoclonal) | Sigma | V9131; RRID:AB_477629 | WB (1:20000) |
| Software, algorithm | GraphPad Prism 7 | GraphPad software | RRID:SCR_002798 | |
| Software, algorithm | kallisto v0.42.4 | *Bray et al., 2016* | RRID:SCR_016582; https://pachterlab.github.io/kallisto/about | |
| Software, algorithm | sleuth v0.30.0 | *Pimentel et al., 2017* | RRID:SCR_016883; https://pachterlab.github.io/sleuth/about | |
| Software, algorithm | featureCounts | *Liao et al., 2014* | RRID:SCR_012919; http://bioinf.wehi.edu.au/featureCounts | |
| Software, algorithm | hisat2 v2.1.0 | *Kim et al., 2019* | RRID:SCR_015530; https://daehwankimlab.github.io/hisat2/ | |
| Software, algorithm | fastx_clipper | Hannon Lab | http://hannonlab.cshl.edu/fastx_toolkit/ | |
| Software, algorithm | fastx_trimmer | Hannon Lab | http://hannonlab.cshl.edu/fastx_toolkit/ | |
| Software, algorithm | bowtie2 v 2.2.3 | *Langmead and Salzberg, 2012* | RRID:SCR_005476; http://bowtie-bio.sourceforge.net/bowtie2/index.shtml | |
| Software, algorithm | RiboWaltz v1.0.1 | *Lauria et al., 2018* | RRID:SCR_016948; https://github.com/LabTranslationalArchitectomics/RiboWaltz | |
| Software, algorithm | DESeq2 v1.22.2 | *Love et al., 2014* | RRID:SCR_015687; https://bioconductor.org/packages/release/bioc/html/DESeq2.html | |
| Software, algorithm | riborex v2.3.4 | *Li et al., 2017* | RRID:SCR_019104; https://github.com/smithlabcode/riborex | |
| Software, algorithm | ensembldb v2.6.8 | *Rainer et al., 2019* | RRID:SCR_019103; https://www.bioconductor.org/packages/release/bioc/html/ensembldb.html | |
| Software, algorithm | Pause site identification algorithm | *Ishimura et al., 2014* | N/A | |
| Software, algorithm | Ingenuity Pathway Analysis (IPA) | QIAGEN, Inc | RRID:SCR_008653; https://www.qiagenbioinformatics.com/products/ingenuity-pathway-analysis | |
| Software, algorithm | DAVID bioinformatics web server | *Huang et al., 2009* | RRID:SCR_001881; http://david.abcc.ncifcrf.gov | |

## Mouse strains

*Gtpbp1*[-/+] mice were generated previously on a mixed genetic background with a portion of exon three and all of exon four replaced with a PGK-Neo cassette (*Senju et al., 2000*). These mice were backcrossed to C57BL/6J mice for more than 10 generations to generate congenic B6J.*Gtpbp1*[-/-] mice and genotyped with the following primers (wild-type forward primer: 5'GAGTACGGGCTGAGTGAAGC3', wild type reverse primer: 5'TGGACAGGAACCTGATGTGA3', mutant forward primer:

5'TACGCCACCGTGAAGAGCAT3', mutant reverse primer: 5'AGGGGAGGAGTGGAAGGTGG3'). Homozygosity for the tRNA (n-Tr20[J/J]) mutation was confirmed by genotyping (*Ishimura et al., 2014*). For transgene rescue experiments, B6J.Tg(n-Tr20[wt]); Gtpbp1[-/-] mice were generated by crossing B6J.Gtpbp1[+/-] mice to B6J-Tg(n-Tr20[wt]) mice that transgenically express wild-type levels of wild type n-Tr20 (*Ishimura et al., 2014*) and then backcrossing to B6J.Gtpbp1[-/+] mice. C57BL/6J (B6J) and B6J.Gcn2[-/-] (Eif2ak4[tm1.2Dron]) mice were obtained from The Jackson Laboratory. Neurological defects were observed in B6J.Gtpbp1[-/-] and B6J-Gtpbp2[-/] males and females; therefore, mice of both sexes were used for experiments. The Jackson Laboratory Animal Care and Use Committee and the University of California San Diego Animal Care and Use Committee approved all mouse protocols (animal protocol number S15286).

## Rapamycin treatment

Rapamycin (LC laboratories) stock solution (50 mg/ml) was prepared in ethanol and diluted on the day of injection in equal volumes of a 10% PEG-400/8% ethanol solution and 10% Tween-80. Mice were injected intraperitoneally with 5 mg/kg rapamycin or vehicle daily; injections were performed from P14-P21 for tissue collection (RNA isolation or western blotting) or from P14-P28 or P28-P42 for histological analysis.

## Histology and immunohistochemistry

Anesthetized mice were transcardially perfused with 4% paraformaldehyde (PFA) for immunofluorescence, 10% neutral buffered formalin (NBF) for in situ hybridization or immunofluorescence, or Bouin's fixative for histology. Tissues were post-fixed overnight and embedded in paraffin. For histological analysis, sections were deparaffinized, rehydrated, and were stained with hematoxylin and eosin according to standard procedures. Histological slides were imaged using a digital slide scanner (Hamamatsu).

For quantification of cerebellar granule cells, the total number of granule cells (viable and pyknotic cells) were counted in a 0.025 mm$^2$ area from lobule VI or IX and averaged from three sections per brain spaced 100 µm apart at midline. For quantification of hippocampal neurons, the number of neurons with, and without, pyknotic nuclei were counted in the CA1 or DG of the hippocampus and averaged from three sections, spaced 50–70 µm apart and about 1.5 mm off midline, per brain. For quantification of pyknotic nuclei in the cortex, the total number of pyknotic nuclei were counted across the entire section and averaged from three sections per animal spaced 50–70 µm apart and about 2–2.5 mm off midline. All histological quantifications were performed with three mice of each genotype and time point using mice of both sexes.

For immunofluorescence, antigen retrieval on deparaffinized PFA-fixed sections was performed by microwaving sections in 0.01M sodium citrate buffer (pH 6.0, 0.05% Tween-20), three times for three minutes each. NBF-fixed sections were microwaved three times for three minutes, followed by two times for nine minutes. PFA- or NBF-fixed sections were incubated with the following primary antibodies: rabbit anti-cleaved caspase 3-D175 (Cell Signaling, #9661, 1:100, PFA-fixed tissue), CTIP2/BCL11B (Abcam, ab18465, 1:500, PFA-fixed tissue), p-S6$^{240/244}$ (Cell Signaling, #5364, 1:1000, NBF-fixed tissue), and RPS6 (Santa Cruz Biotechnology, sc-74459, 1:500, NBF-fixed tissue). Detection of primary antibodies was performed with goat anti-mouse Alexa Fluor-488, goat anti-rabbit Alexa Fluor-488, goat anti-rat Alexa Fluor-555, and goat anti-rabbit Alexa Fluor-555 secondary antibodies (Invitrogen). Sections were counterstained with DAPI, and treated with Sudan black to quench autofluorescence.

## RNAscope (in situ hybridization)

In situ hybridization of Gtpbp1, Gtpbp2, Sesn2, Slc7a1, Ddr2 and Chac1 probes was performed as per the manufacturer's protocol (RNAscope Multiplex Fluorescent Reagent Kit v2; Advanced Cell Diagnostics). Briefly, deparaffinized NBF-fixed sections were treated for 15 min with Target Retrieval Reagent at 100 ℃ and subsequently treated with Protease Plus for 30 min at 40 ℃. RNAscope probes were hybridized for 2 hr. TSA Plus Cyanine 5 (PerkinElmer, 1:1000) was used as a secondary fluorophore for C1 probes (Gtpbp2, Slc7a1, Chac1) and TSA Plus Cyanine 3 (PerkinElmer, 1:2000) was used as a secondary fluorophore for C2 probes (Gtpbp1, Sesn2, Ddr2).

## Reverse transcription, quantitative PCR, and genomic PCR analysis

Cerebella or hippocampi were isolated and immediately frozen in liquid nitrogen. Total RNA was extracted with Trizol reagent (Life Technologies). cDNA synthesis was performed on DNase-treated (DNA-free DNA Removal Kit, Life Technologies AM1906) total RNA using oligo(dT) primers and the SuperScript III First-Strand Synthesis System (Life Technologies). Quantitative RT-PCR (qRT) reactions were performed using iQ SYBR Green Supermix (Bio-Rad) and a CFX96 Real-Time PCR Detection System (Bio-Rad). Reactions were performed with primers previously published (*Ishimura et al., 2016*). Expression levels of the genes of interest were normalized to *β-actin (Actb)* (forward primer: 5'GGCTGTATTCCCCTCCATCG3', reverse primer: 5' CCAGTTGGTAACAATGCCATGT3') using the $2^{-\Delta\Delta CT}$ method (*Livak and Schmittgen, 2001*) and expressed as fold change + standard error of the mean (SEM) relative to control (B6J).

## RNA-Seq library preparation

Cerebella from various strains were isolated from 3-week-old mice (P21, n = 3 mice for each genotype) and immediately frozen in liquid nitrogen. RNA-Seq libraries were prepared using the TruSeq v2 mRNA kit (Illumina). Briefly, total cerebellar RNA was isolated and RNA quality was assessed on an Agilent TapeStation. mRNA was purified using biotin-tagged poly(dT) oligonucleotides and streptavidin-coated magnetic beads. After fragmentation of mRNA, libraries were prepared according to the manufacturer's instructions. Paired end reads (2 × 100 bp) were obtained using the HiSeq 4000 (Illumina).

## Ribosome profiling library construction

Ribosome profiling libraries were generated as previously described (*Ingolia et al., 2012*; *Ishimura et al., 2014*) with some minor modifications. Briefly, dissected cerebella were immediately frozen in liquid nitrogen. One cerebellum from P21 mice was used for each biological replicate, and three biological replicates were prepared for each genotype. Tissue homogenization was performed with a mixer mill (Retsch MM400) in 350 μl lysis buffer (20 mM Tris-Cl, pH 8.0, 150 mM NaCl, 5 mM MgCl$_2$, 1 mM DTT, 100 μg/ml CHX, 1% (v/v) TritonX-100, 25 units/ml Turbo DNaseI). Lysates were treated with RNase I and overlaid on top of a sucrose cushion in 5 ml Beckman Ultraclear tubes and centrifuged in an SW55Ti rotor for 4 hr at 4°C at 46,700 rpm. Pellets were resuspended and RNA was extracted using the miRNeasy kit (QIAGEN) according to the manufacturer's instructions. 26–34 nucleotide RNA fragments were purified by electrophoresis on a 15% denaturing polyacrylamide gel. Linker addition, cDNA generation (first-strand synthesis was performed at 50°C for 1 hr), circularization, rRNA depletion, and amplification of cDNAs with indexing primers were performed. Library quality and concentration were assessed using High Sensitivity D1000 ScreenTapes on the Agilent TapeStation, Qubit 2.0 Fluorometer, and qPCR. Libraries were run on a HiSeq4000 (SR75).

## RNA-Seq data analysis

Reads were quantified using kallisto version 0.42.4 (*Bray et al., 2016*) and pseudo-aligned to a Gencode M20 transcriptome reference with parameters –bias and -b 100. Differential expression was performed using sleuth version 0.30.0 (*Pimentel et al., 2017*). To identify differentially expressed genes the following pairwise comparisons were performed: B6J vs. B6J.*Gtpbp1*$^{-/-}$, B6J vs. B6J-*Gtpbp2*$^{-/-}$, and B6J.*Gtpbp1*$^{-/-}$ vs. B6J-*Gtpbp2*$^{-/-}$. Using functions within sleuth, we aggregated transcript expression at the gene level (by Ensembl gene identifier), fit null models and models corresponding to the genotype of the samples for each gene, and performed Wald tests on the models for each gene to identify differentially expressed genes. Multiple hypothesis testing correction was done using a Benjamini-Hochberg correction, which is the default method in sleuth and referred to as q-value. For downstream TE analysis, mapping to mm10 using a Gencode M20 transcript annotation was performed using hisat2 version 2.1.0 (*Kim et al., 2019*) using default parameters.

## Ribosome profiling data analysis

Reads were clipped to remove adaptor sequences (CTGTAGGCACCATCAAT) using fastx_clipper and trimmed so that reads start on the second base using fastx_trimmer (http://hannonlab.cshl.edu/fastx_toolkit/). Reads containing ribosomal RNA were then filtered out by mapping to a ribosomal RNA reference using bowtie2 version 2.2.3 using parameters -L 13 (*Langmead and Salzberg, 2012*).

Remaining reads were mapped to a mm10 mouse reference using a Gencode M20 annotation, or a Gencode M20 transcript reference using hisat2 version 2.1.0 (*Kim et al., 2019*). Ribosomal A-sites were identified using RiboWaltz version 1.0.1 (*Lauria et al., 2018*), and read lengths 29–33 were retained for further analysis. Observed/expected reads were calculated with the expected reads being the read density expected at a given site with a given codon, assuming that reads are uniformly distributed across the coding part of the transcript. Pauses were identified using previous methodology (*Ishimura et al., 2014*) using a 0.5 reads/codon in all samples to threshold transcripts to analyze. Correlation between gene pauses and gene expression was calculated by taking the average pause scores of transcripts associated with a gene for transcripts with $\geq$0.5 reads/codon across the B6J.*Gtpbp1*$^{-/-}$ samples. The sum of these pause scores across all replicates was then correlated with the sleuth *beta* values (in the B6J.*Gtpbp1*$^{-/-}$ vs. B6J comparison) for genes that passed the reads/codon thresholding in the ribosome profiling datasets.

Read counts for translational efficiency (TE) analysis were quantified using featureCounts (*Liao et al., 2014*) with footprints overlapping CDS features and RNA read pairs overlapping gene exon features. Histone mRNAs were removed from the analysis by removing gene names with the prefix 'Hist' and filtering out genes in HistoneDB 2.0 (*Draizen et al., 2016*). Differential TE analysis was performed using riborex version 2.3.4 using the DESeq2 engine (*Li et al., 2017*). TE analysis of AGA A-site filtered datasets was performed by identifying AGA codons in the transcriptome and transferring the coordinates to the genome using ensembldb v.2.6.8 (*Rainer et al., 2019*). Then reads with AGAs in the A-site (±1 codon based on riboWaltz A-site offset for each read length) were removed and the above TE analysis was performed. A list of mouse genes with known TOP motifs were identified in *Yamashita et al., 2008*.

## Gene ontology (GO) and pathway analysis

RNA-sequencing data were analyzed using Ingenuity Pathway Analysis (QIAGEN Inc, https://www.qiagenbioinformatics.com/products/ingenuity-pathway-analysis). Gene Ontology (GO) pathway analysis was performed using the DAVID bioinformatics web server (http://david.abcc.ncifcrf.gov) by uploading the gene lists from our ribosome profiling analysis (AGA pausing genes, z $\geq$ 10, stalls detected in any biological replicates). The functional annotation chart and clustering analysis modules were utilized for gene-term enrichment analysis, and terms with a Benjamini-Hochberg adjusted p-value$\leq$0.05 were considered enriched.

## Western blotting

Cerebella or hippocampi were isolated and immediately frozen in liquid nitrogen. Proteins were extracted by homogenizing tissue in 5 volumes of RIPA buffer with cOmplete Mini, EDTA-free Protease Inhibitor Cocktail (Roche), sonicated two times for 10 s (Branson, 35% amplitude), incubated for 30 min at 4°C, centrifuged at 16,000xg for 25 min, and 25 µg of whole protein lysate were resolved on SDS-PAGE gels prior to transfer to PVDF membranes (GE Healthcare Life Sciences, #10600023) using a tank blotting apparatus (BioRad).

For detection of phosphoproteins, frozen tissue samples were homogenized in 5 volumes of homogenization buffer (50 mM HEPES/KOH, pH 7.5, 140 mM potassium acetate, 4 mM magnesium acetate, 2.5 mM dithiothreitol, 0.32M sucrose, 1 mM EDTA, 2 mM EGTA) (*Carnevalli et al., 2004*), supplemented with phosphatase and protease inhibitors (PhosStop and cOmplete Mini, EDTA-free Protease Inhibitor Cocktail, Roche). Lysate samples were centrifuged at 12,000xg for 7 min, and either 30 µg (detection of ribosomal protein S6) or 70 µg (detection of eIF2alpha) of whole protein lysate were resolved on SDS-PAGE gels prior to transfer to PVDF membranes. After blocking in 5% nonfat dry milk (Cell Signaling, #9999S), blots were probed with primary antibodies at 4°C overnight: rabbit anti-phospho-S6$^{240/244}$ (Cell Signaling, #5364, 1:4000), mouse anti-RPS6 (Santa Cruz Biotechnology, sc-74459, 1:2000), rabbit anti-phospho-eIF2alpha$^{S51}$ (Cell Signaling, #9721, 1:1000), rabbit anti-eIF2alpha (Cell Signaling, #9722, 1:2000), mouse anti-vinculin (Sigma, V-9131, 1:20,000). Followed by incubation with HRP-conjugated secondary antibodies for 2 hr at room temperature: goat anti-rabbit IgG (BioRad, #170–6515) or goat anti-mouse IgG (BioRad, #170–6516). Signals were detected with SuperSignal West Pico Chemiluminescent Substrate (ThermoScientific, #34080).

## Statistics

For quantification of protein expression, RNA expression, or histological quantifications, p-values were computed in GraphPad Prism using either multiple *t*-tests, one-way ANOVA, or two-way ANOVA, and corrected for multiple comparisons as indicated in the figure legends. All quantifications were performed with at least three mice of each genotype and time point. Neurological defects were observed in B6J.*Gtpbp1*$^{-/-}$ and B6J-*Gtpbp2*$^{-/-}$ male and female mice. Therefore, mice of both sexes were used for experiments.

## Acknowledgements

We thank T Jucius, and A Kano for technical assistance, Drs. Satori Senju and Yasuharu Mishimura for providing *Gtpbp1*$^{-/-}$ mice, the IGM Genomics Center at the University of California San Diego (UCSD) for support with the preparation of RNA-sequencing libraries and sequencing, and the UCSD School of Medicine Microcopy Core for providing access to microscopy equipment (Grant P30 NS047101). This work was supported by NIH NS094637 (SLA). SLA is an investigator of the Howard Hughes Medical Institute.

## Additional information

### Competing interests

Susan L Ackerman: Is a Reviewing Editor for eLife. The other authors declare that no competing interests exist.

### Funding

| Funder | Grant reference number | Author |
| --- | --- | --- |
| National Institute of Neurological Disorders and Stroke | NS094637 | Susan L Ackerman |
| Howard Hughes Medical Institute | | Susan L Ackerman |

The funders had no role in study design, data collection and interpretation, or the decision to submit the work for publication.

### Author contributions

Markus Terrey, Conceptualization, Formal analysis, Investigation, Methodology, Writing - original draft, Writing - review and editing, Designed experiments and wrote the paper. Performed experiments using *Gcn2*-/- mice. Performed mouse and molecular biology experiments under SLA's guidance; Scott I Adamson, Formal analysis, Investigation, Methodology, Writing - review and editing, Performed the computational analysis of RNA-sequencing and ribosome profiling data under JHC's guidance; Alana L Gibson, Investigation, Performed RNAscope and immunofluorescence experiments for ribosomal protein S6; Tianda Deng, Investigation, Performed experiments using *Gcn2*-/- mice; Ryuta Ishimura, Investigation, Performed initial phenotype analysis of Gtpbp1-/- mice; Jeffrey H Chuang, Supervision, Writing - review and editing; Susan L Ackerman, Conceptualization, Supervision, Funding acquisition, Writing - review and editing, Designed experiments and wrote the paper

### Author ORCIDs

Alana L Gibson http://orcid.org/0000-0003-2247-7064
Susan L Ackerman https://orcid.org/0000-0002-6740-593X

### Ethics

Animal experimentation: The Jackson Laboratory Animal Care and Use Committee and the University of California San Diego Animal Care and Use Committee approved all mouse protocols. Animal protocol number S15286.

Decision letter and Author response
Decision letter https://doi.org/10.7554/eLife.62731.sa1
Author response https://doi.org/10.7554/eLife.62731.sa2

## Additional files

### Supplementary files

• Supplementary file 1. All AGA pausing genes with increased ribosome occupancy (z-score ≥10) observed in P21 B6J.*Gtpbp1*$^{-/-}$ (sheet 1) and B6J-*Gtpbp2*$^{-/-}$ (sheet 2) mice (n = 3 mice/genotype).

• Supplementary file 2. Transcriptional gene expression analysis in the cerebellum of P21 B6J and B6J.*Gtpbp1*$^{-/-}$ (sheet 1), B6J and B6J-*Gtpbp2*$^{-/-}$ (sheet 2), or B6J.*Gtpbp1*$^{-/-}$ and B6J-*Gtpbp2*$^{-/-}$ (sheet 3) mice (n = 3 mice/genotype).

• Supplementary file 3. Analysis of differential translation efficiency in the cerebellum of P21 B6J and B6J.*Gtpbp1*$^{-/-}$ (sheet 1) or B6J and B6J-*Gtpbp2*$^{-/-}$ (sheet 2) mice (n = 3 mice/genotype).

• Transparent reporting form

### Data availability

All data generated or analysed during this study are included in the manuscript and supporting files. The RNA sequencing and ribosome footprint data have been made available (GSE157902).

The following dataset was generated:

| Author(s) | Year | Dataset title | Dataset URL | Database and Identifier |
|---|---|---|---|---|
| Terrey M, Adamson SI, Gibson AL, Deng T, Ishimura R, Chuang JH, Ackerman SL | 2020 | GTPBP1 resolves paused ribosomes to maintain neuronal homeostasis | https://www.ncbi.nlm.nih.gov/geo/query/acc.cgi?acc=GSE157902 | NCBI Gene Expression Omnibus, GSE157902 |

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
