## [Decision Letter]

Thank you for submitting your article "GTPBP1 resolves paused ribosomes to maintain neuronal homeostasis" for consideration by *eLife*. Your article has been reviewed by three peer reviewers, including David Ron as the Reviewing Editor and Reviewer #1, and the evaluation has been overseen by Huda Zoghbi as the Senior Editor. The following individual involved in review of your submission has agreed to reveal their identity: Nahum Sonenberg (Reviewer #3).

The reviewers have discussed the reviews with one another and the Reviewing Editor has drafted this decision to help you prepare a revised submission.

Summary:

This manuscript addresses the functions of translational GTPases in the resolution of stalling of elongating ribosomes and neuronal homeostasis. Prior studies from the same laboratory suggested that GTPBP2 is essential for release of ribosomes stalled at Arg codons; loss of GTPBP2 increased ribosome stalling, activated *Gcn2* and the Integrated stress response (ISR), and enhanced neurodegeneration. The present manuscript uses elegant mouse genetic models and associated cell biology and biochemistry to explore similar questions with the related GTPBP1. Key discoveries are that loss of GTPBP1 leads to similar ribosome stalling and degeneration in selected neuronal tissues. Furthermore, loss of GTPBP1 activates GCN2 and loss of this eIF2 kinase accelerates degeneration of brain sections. Importantly, the compound GTPBP1/2 mutant is no more impaired than either single mutation. These results indicate while GTPBP1 and GTPBP2 are highly related they are not redundant, rather they function at different steps in the same pathway. The manuscript shows that there are differences in mTORC1 signaling among the neuronal tissues upon loss of GTPBP1. mTORC1 signaling and the integrated stress response are inversely regulated in response to stress and the loss of GTPBP1 and associated ribosome stalling is linked with lowered mTORC1 in hippocampus, whereas mTORC1 signaling was not altered in granule cells, comprising most of the cerebellum. Therefore, there can be concordant and discordant regulation between the integrated stress response and mTORC1 depending on the neuronal cell types; pharmacological inhibition of mTORC1 upon GTPBP1 depletion enhanced cell death.

Overall, this is a significant and timely study that would be of broad interest. The manuscript provides new insights into resolution of ribosome stalling, associated regulation of stress response pathways, and mechanisms underlying neurodegeneration and supports the surprising conclusion that despite their similarity GTPBP1 and GTPBP2 have different functions in the same pathway. Experiments are rigorous and appropriately interpreted. The manuscript is clearly presented and flows logically. As you will read in the unedited reviews the reviewers initially differed in their assessment of the significance of the findings relating to TOR, but the consensus to emerge was to favour the less critical view of reviewers 2 and 3, on this point.

Essential revisions:

Listed below are the points we find especially important to address in a revised version of the manuscript, that we hope to receive from you in short order.

1) The manuscript features ribosome profiling and RNA-seq that are presented sequentially in the subsection “GTPBP1 is a novel ribosome-rescue factor”. However, which dataset is being analyzed and discussed is not always clear to the reader. It should be clarified whether the profiling was normalized for RNA- translational efficiencies. This normalization point should be clarified in the Results text and the legend for Figure 2. Figure 2D refers to analysis of differential gene expression, a generic term that is used later for the RNA-seq analysis. More precision in the language through the translational efficiency and mRNA levels would be helpful. Furthermore, Figure 3 should clarify whether the "analysis of differential gene expression" refers to mRNA measurements.

2) At several points in the manuscript it is stated that ribosome stalling induces GCN2 and mTORC1 signaling. The evidence for GCN2 induction by ribosome stalling is strong based on research from this lab and others. The idea that ribosome stalling directly induces neuron-specific changes in mTOR signaling warrants some caution (as opposed to indirect regulation through changes in the integrated stress response, amino acid pools, or other mechanisms).

3) GTPBP1 and GTPBP2 are very similar and act by a similar mechanism. It is thus very surprising that only 50% of AGA pausing genes with increased AGA occupancy intersected between B6J.*Gtpbp1^-/-^* and B6J-*Gtpbp2^-/-^* mice (Figure 2—figure supplement 1B, Supplementary file 1). It is even more surprising that the biological processes as defined by Gene Ontology (GO) analysis of pausing genes

revealed enrichment in different biological processes between *Gtpbp1* and *Gtpbp2* (Figure 2—figure supplement 1C). All this consistent with the finding that GTPBP1 and GTPBP2 are not mechanistically redundant, but the mechanistic insight, although not known should be mentioned and discussed.

4) Figure 1C – It appears that there is much more GTPBP2 than GTPBP1. Is it correct? What is the ratio? Also, the two proteins do not co-localize. Localization is even mutually exclusive. What do these results mean?

5) Figure 3C – why is the amount of S6 and its phosphorylated form reduced in the hippocampus, but not in the cerebellum. Any differences in mTOR or upstream effectors amounts between these two structures?

6) The experimental evidence provided here strongly supports the conclusion that the two GTPBP's function in the same pathway. This might be expected, given the similarity of the encoded proteins but surprisingly, the compound BL6/J GTPBP1∆/∆;GTPBP2∆/∆ is no more compromised than either BL6/J GTPBP1∆/∆ or BL6/J GTPBP2∆/∆ single mutants. This last observation provides genetic evidence that despite considerable similarity in their encoded proteins, the two genes act at different steps in the same pathway. This is a surprising finding that stands to influence thinking in this area and will generate a search for the mechanistic basis of differences in function between GTPBP1 and GTPBP2 and the specific steps they act on in coping with stalled ribosomes (see also point 3, above). Less interesting would be a study that concluded that these two proteins do more or less the same thing and that the gene duplication that gave rise to their presence in the mammalian genome represents some sort of fine tuning, the basis of which remains for now obscure. The critical experiment examining this point is shown in Figure 1F and G, which compare the number of granule cells in two lobes of the cerebellum at a single time point (6 weeks) in three mice each with the relevant genotypes. The equivalence of the loss of cells between the three relevant genotypes (GTPBP1∆/∆;GTPBP2∆/∆, GTPBP1∆/∆ and GTPBP2∆/∆) is impressive and fits well the conclusion that GTPBP1 and GTPBP2 are active on different steps in the same pathway. However, the lack of a temporal component to the measurements, limits the strength of that key conclusion. For example, it is possible that by six weeks of age the histological changes had plateaued, obscuring important differences in the rate at which that plateau had been reached by the different genotypes. If the compound mutants were actually to degenerate more quickly than either single mutant, the paper's key conclusion would be wrong in that it would suggest that the two GTPBP's are in fact redundant and carry out the same step biochemically. Addressing this issue seems key to the significance of this paper.

---

## [Author Response]

Essential revisions:Listed below are the points we find especially important to address in a revised version of the manuscript, that we hope to receive from you in short order.1) The manuscript features ribosome profiling and RNA-seq that are presented sequentially in the subsection “GTPBP1 is a novel ribosome-rescue factor”. However, which dataset is being analyzed and discussed is not always clear to the reader. It should be clarified whether the profiling was normalized for RNA- translational efficiencies. This normalization point should be clarified in the Results text and the legend for Figure 2. Figure 2D refers to analysis of differential gene expression, a generic term that is used later for the RNA-seq analysis. More precision in the language through the translational efficiency and mRNA levels would be helpful. Furthermore, Figure 3 should clarify whether the "analysis of differential gene expression" refers to mRNA measurements.

We appreciate the input as this may also confuse readers. We have clarified our language to specifically identify the datasets we are referring to. In addition, we defined in the text, legends, and Materials and methods, that translation efficiency (TE) data reflect normalization of the abundance of ribosome-protected fragments to that of RNA-Seq reads and refer to these data more consistently as TE *Gtpbp1* or TE *Gtpbp2*. Our RNA-Seq data were utilized to measure mRNA abundance and we now refer to changes in mRNA abundance as transcriptional differences in gene expression (DE *Gtpbp1* or DE *Gtpbp2*) in the text and in the legend for Figure 3.

2) At several points in the manuscript it is stated that ribosome stalling induces GCN2 and mTORC1 signaling. The evidence for GCN2 induction by ribosome stalling is strong based on research from this lab and others. The idea that ribosome stalling directly induces neuron-specific changes in mTOR signaling warrants some caution (as opposed to indirect regulation through changes in the integrated stress response, amino acid pools, or other mechanisms).

We agree with the reviewer and did not mean to imply that changes in mTOR signaling are a direct result of ribosome stalling. We have included a statement in the Discussion to this effect.

3) GTPBP1 and GTPBP2 are very similar and act by a similar mechanism. It is thus very surprising that only 50% of AGA pausing genes with increased AGA occupancy intersected between B6J.Gtpbp1^-/-^ and B6J-Gtpbp2^-/-^ mice (Figure 2—figure supplement 1B, Supplementary file 1). It is even more surprising that the biological processes as defined by Gene Ontology (GO) analysis of pausing genesrevealed enrichment in different biological processes between Gtpbp1 and Gtpbp2 (Figure 2—figure supplement 1C).

Pausing on AGA codons is very robust in the *Gtpbp1^-/-^* cerebellum as shown in Figure 2 A and B, and is consistent with our previous data from the *Gtpbp2^-/-^* cerebellum. However, the genes with pause sites in each biological replicate for either genotype only overlap by about 50% (Figure 2C). This suggests that ribosome pausing is stochastic, i.e., it can occur at any AGA codon in any gene, but doesn’t occur at all codons at a given time. Capturing all stochastic events would require a substantially higher sequencing depth and therefore, we likely only scratched the surface in defining pausing events in these mice. Furthermore, as pointed out in the Results and the Materials and methods, we only considered strong pauses for our analysis (z-score above 10, that is pausing is at least 10 standard deviations above background). Our thresholds are picked to have high confidence in our pausing data. However, weaker pauses may be missed at the expense of high confidence. Thus, all things considered, we wouldn’t expect to observe a higher overlap between two different genotypes.

As we would expect from stochastic pausing, pausing genes are enriched for many GO terms. Most of TOP GO terms (17 out of 24) overlap between *Gtpbp1* and *Gtpbp2* and the GO terms that appear unique to pauses which occur in a given genotype, are again likely due to the depth of our ribosome profiling datasets. Indeed, to circumvent the limitations of single/individual experiments, our conclusion of *Gtpbp1* and *Gtpbp2* are likely acting in the same pathway is not just derived from our profiling data, but rather is a collective hypothesis based on both our genomic and genetic data.

All this consistent with the finding that GTPBP1 and GTPBP2 are not mechanistically redundant, but the mechanistic insight, although not known should be mentioned and discussed.

Our study, together with the recent biochemical study from Zinoviev et al., 2018, are the first studies that analyze *Gtpbp1* and *Gtpbp2* side by side. Thus, we only have limited understanding about the shared or unique functions of these two proteins and we have included a statement in the Discussion to point out that the function of these proteins is still unknown. The biochemical experiments by Zinoviev et al. revealed striking differences in mRNA decay for these two proteins which we have discussed in the Discussion.

4) Figure 1C – It appears that there is much more GTPBP2 than GTPBP1. Is it correct? What is the ratio? Also, the two proteins do not co-localize. Localization is even mutually exclusive. What do these results mean?

Figure 1C shows our in situ hybridization experiments for mRNA as mentioned in the Results and figure legend. Thus, the fluorescent punta do not indicate protein localization, nor does the nature of this experiment (different probes for each gene that are labeled with different fluorophores) allow us to measure levels of *Gtpbp1* and *Gtpbp2* transcripts. However, these experiments do show that the mRNAs for *Gtpbp1* and *Gtpbp2* are found in the same neurons and are expressed in both neurons that degenerate and those that do not.

5) Figure 3C – why is the amount of S6 and its phosphorylated form reduced in the hippocampus, but not in the cerebellum. Any differences in mTOR or upstream effectors amounts between these two structures?

We do not know why the mTOR pathway responds to defects in translation in cell type-specific fashion. Both positive and negative regulators of the mTOR pathway (e.g. AKT, ERK, PTEN, TSC1/2, etc.) are ubiquitously expressed (as far as we know) and we detected expression these genes in our transcriptome datasets from the cerebellum. However, proteomics data might be more insightful as many mTOR regulators act at the protein level and/or thru post-translational modifications. Even so to point to differences in upstream effectors that alter mTOR would require systemic genetic and/or molecular studies across different neuronal populations.

A large body of literature has shown that the mTOR pathway can respond to numerous cellular insults and that the activity of this signaling pathway is influenced by changes in cellular homeostasis including protein synthesis, neuronal activity and energy balance. The particular alteration(s) in cellular homeostasis in the *Gtpbp1^-/-^* and *Gtpbp2^-/-^* brain that triggers a decrease in mTOR activity is unknown, making it even more difficult to speculate about the cell type-specific responses we observed. In the Discussion, we suggest that defects in translation in different cell types lead to differences in the amount and type of perturbations, perhaps due to the differences in basal translation between cells, and this in turn could result in differential homeostatic changes.

6) The experimental evidence provided here strongly supports the conclusion that the two GTPBP's function in the same pathway. This might be expected, given the similarity of the encoded proteins but surprisingly, the compound BL6/J GTPBP1∆/∆;GTPBP2∆/∆ is no more compromised than either BL6/J GTPBP1∆/∆ or BL6/J GTPBP2∆/∆ single mutants. This last observation provides genetic evidence that despite considerable similarity in their encoded proteins, the two genes act at different steps in the same pathway. This is a surprising finding that stands to influence thinking in this area and will generate a search for the mechanistic basis of differences in function between GTPBP1 and GTPBP2 and the specific steps they act on in coping with stalled ribosomes (see also point 3, above). Less interesting would be a study that concluded that these two proteins do more or less the same thing and that the gene duplication that gave rise to their presence in the mammalian genome represents some sort of fine tuning, the basis of which remains for now obscure. The critical experiment examining this point is shown in Figure 1F and G, which compare the number of granule cells in two lobes of the cerebellum at a single time point (6 weeks) in three mice each with the relevant genotypes. The equivalence of the loss of cells between the three relevant genotypes (GTPBP1∆/∆;GTPBP2∆/∆, GTPBP1∆/∆ and GTPBP2∆/∆) is impressive and fits well the conclusion that GTPBP1 and GTPBP2 are active on different steps in the same pathway. However, the lack of a temporal component to the measurements, limits the strength of that key conclusion. For example, it is possible that by six weeks of age the histological changes had plateaued, obscuring important differences in the rate at which that plateau had been reached by the different genotypes. If the compound mutants were actually to degenerate more quickly than either single mutant, the paper's key conclusion would be wrong in that it would suggest that the two GTPBP's are in fact redundant and carry out the same step biochemically. Addressing this issue seems key to the significance of this paper.

We included additional time points (3 and 4 weeks) in Figure 1 for the double mutants. Our data show that onset and progression of granule cell death are very similar between *Gtpbp1^-/-^*, *Gtpbp2^-/-^*, and *Gtpbp1^-/-^*; *Gtpbp2^-/-^* cerebella.